# Inequality in Health: The Correlation between Poverty and Injury—A Comprehensive Analysis Based on Income Level in Taiwan: A Cross-Sectional Study

**DOI:** 10.3390/healthcare9030349

**Published:** 2021-03-18

**Authors:** Shi-Hao Huang, Shih-Chun Hsing, Chien-An Sun, Chi-Hsiang Chung, Chang-Huei Tsao, Ren-Jei Chung, Bing-Long Wang, Yao-Ching Huang, Wu-Chien Chien

**Affiliations:** 1Department of Chemical Engineering and Biotechnology, National Taipei University of Technology (Taipei Tech), Taipei 10608, Taiwan; hklu2361@gmail.com (S.-H.H.); ph870059@gmail.com (Y.-C.H.); 2Center for Healthcare Quality Management, Cheng Hsin General Hospital, Taipei 11220, Taiwan; ch8363@chgh.org.tw; 3Department of Health Care Management, College of Health Technology, National Taipei University of Nursing and Heath Sciences, Taipei 10608, Taiwan; 4Department of Public Health, College of Medicine, Fu-Jen Catholic University, New Taipei 24205, Taiwan; 040866@mail.fju.edu.tw; 5Big Data Research Center, College of Medicine, Fu-Jen Catholic University, New Taipei 24205, Taiwan; 6School of Public Health, National Defense Medical Center, Taipei 11490, Taiwan; g694810042@gmail.com (C.-H.C.); billwang1203@gmail.com (B.-L.W.); 7Department of Medical Research, National Defense Medical Center, Tri-Service General Hospital, Taipei 11490, Taiwan; changhuei@mail.ndmctsgh.edu.tw; 8Department of Microbiology & Immunology, National Defense Medical Center, Taipei 11490, Taiwan; 9The Graduate Institute of Life Sciences, National Defense Medical Center, Taipei 11490, Taiwan

**Keywords:** health inequality, health care, low income, nonlow income, injured inpatient

## Abstract

Is income still an obstacle that influences health in Taiwan, the National Health Insurance system was instituted in 1995? After collecting injured inpatient data from the health insurance information of nearly the whole population, we categorized the cases as either low-income or nonlow-income and tried to determine the correlation between poverty and injury. Chi-square tests, Fisher’s exact tests, an independent-samples t-test, and percentages were used to identify differences in demographics, causes for hospitalization, and other hospital care variables. Between 1998 and 2015, there were 74,337 inpatients with low-income injuries, which represented 1.6% of all inpatients with injury events. The hospitalization mortality rate for the low-income group was 1.9 times higher than that of the nonlow-income group. Furthermore, the average length of hospital stay (9.9 days), average medical expenses (1681 USD), and mortality rate (3.6%) values for the low-income inpatients were higher than those of the nonlow-income group (7.6 days, 1573 USD, and 2.1%, respectively). Among the injury causes, the percentages of “fall,” “suicide,” and “homicide” incidences were higher for the low-income group than for the nonlow-income group. These findings support our hypothesis that there is a correlation between poverty and injury level, which results in health inequality. Achieving healthcare equality may require collaboration between the government and private and nonprofit organizations to increase the awareness of this phenomenon.

## 1. Introduction

Measuring health inequalities requires information about individual-level health and socioeconomic status. Some studies use individual-level information to explore the degree of inequality and the causal relationship between socioeconomic status and health inequality. Previous research has shown that there are income-related health inequalities across Europe and that income inequality has a significant impact on health inequality [1]. Mangalore used a concentration index to measure income-related inequality in British mental health and found that obvious inequality is detrimental to low-income groups [2]. A study using national data on elderly Chinese individuals found that not only personal income but also provincial income affects their health [3].

In 1995, Taiwan instituted the National Health Insurance (NHI) system, which requires mandatory participation for all citizens from birth and provides a basic level of medical care for all citizens. The NHI system greatly increased the accessibility of medical care, reduced the financial obstacles to obtaining medical services, prevented financial hardship caused by the cost of medical care, and encouraged those without money to seek medical care [4].

Previous research has revealed that while low-income individuals comprise 0.7% of the total population, their inpatient care claims represent 4.8% of national inpatient costs. Additionally, although only 4.3% of the low-income population receives social welfare assistance, their NHI expenses account for 14.4% of the total NHI costs [5]. Other research has shown that hospitalization rates for low-income individuals (14.1%) are 8.5% higher than those of nonlow-income individuals, indicating that the low-income population utilizes more medical resources than the national average [6].

Health is an important part of human capital. Maintaining economic growth and improving health are based solely on economic reasons. Good health improves the level of human capital, which has a positive impact on personal productivity and human capital returns. Better sanitation improves the productivity of the workforce by reducing the number of days lost to work and sick leave, and it also increases the chances of obtaining high-paying jobs. Although health can be seen as a form of human capital that has a beneficial effect on productivity, income can also affect health in a positive way. The ability to generate higher income helps increase the consumption of health-related commodities—such as adequate food and medicine—and the use of healthcare, which promotes a longer lifespan. However, the impact of income on health is unevenly distributed. The wealthier can provide higher investments in health capital, although the marginal benefits are the largest among the poorest. Therefore, it is necessary to analyze the health–income–injury relationship in a framework where income inequality also affects health outcomes [7].

However, previous research has focused on either low-income families without a control group or on overall medical care expenses rather than medical care costs associated solely with accidental injuries. Therefore, the purpose of our research was to compare the characteristics of “low-income” and “nonlow-income” injury inpatients and evaluate correlations between income level and health inequality. The main hypothesis was concerned with whether poverty correlates with injury. We hypothesized that there is a correlation between poverty and injury level, which results in health inequality.

## 2. Materials and Methods

### 2.1. Data Source

Implemented in 1995, the NHI system currently covers 99% of all Taiwanese citizens. The Health and Welfare Data Science Center, Ministry of Health and Welfare (HWDC, MOHW), collects all emergency room and hospitalization data. Furthermore, the law requires medical facilities to submit claims for emergency rooms and hospitalization expenses on a monthly basis. Therefore, the HWDC is the most authoritative data source for medical and healthcare-related research [8]. This study utilized a population-based, cross-sectional design. We used the original inpatient and outpatient medical claims data collected between 1998 and 2015 (TSGHIRB number 1-105-05-142).

### 2.2. Variable Definitions

The variables included the following: low income (yes, no), gender (male, female), age (1–4, 5–14, 15–24, 25–44, 45–64, and ≥65 years), Charlson comorbidity index (CCI), intentionality of injury (ICD-9-CM E-Code: E800–E949—unintentional; E950–E979—intentional; and E980–E989—unspecific and unable to determine), cause of injury (ICD-9-CM E-Code: E800–E848—transport-related injuries; E850–E869—poisoning; E870–E879—medical malpractice; E880–E888—falls; E890–E899—burns; E900–E909—natural and environmental factors; E910—drowning; E911–E915—suffocation; E916—E920 crushing; cutting and piercing; E921–E949—others unintentional; E950–E959—suicide; E960–E979—homicide; E980–E989—undetermined), surgical operation (yes, no), level of care (medical center, regional hospital, local hospital), hospitalization area (northern, central, southern, eastern, outer islands), length of hospital stay (days), medical expense (USD), and prognosis (survival, mortality). The low-income qualification was stipulated by Article 4 of the Public Assistance Act of Taiwan with the following conditions: (1) individuals must submit an application and be approved by their local municipality authority, (2) the average monthly income per person in the household must fall below the poverty line, and (3) the total household assets must not exceed the specific amount set by the central and municipality authorities in the year the application is submitted. The poverty line was based on the standard published by the Central Department of Budget, Accounting, and Statistics, and is defined by the central and municipality authorities as 60% of the median personal expenditure amount in the household’s local area in the past year [9].

The CCI [10] selects the first five diagnostic codes (ICD-9-CM N-Code), weighs them according to scoring criteria defined by Charlson, and calculates the total score. Higher scores indicate more complications or a more severe diagnosis. Additionally, the “prognosis” for the deceased included deaths in the hospital and voluntary discharge for the terminally ill.

### 2.3. Statistical Analysis

This study utilized a population-based, cross-sectional design. IBM SPSS Statistics 20.0 was used to conduct all statistical analyses. Statistical significance was set at *p* < 0.05. A binary variable denoting a person’s status as poor (or nonpoor) or a variable denoting the deprivation score was assigned to those who were considered to be poor. Univariate statistics and multivariate logistic regression were used to compare mortality rates during hospitalization between the low-income and nonlow-income groups, with survival acting as the dependent variable (surviving or deceased); and demographics, hospitalization cause, and other hospitalization care measures serving as the independent variables. Through this multivariate logistic regression analysis, we could partially study these transmission mechanisms by studying the determinants of poverty. In the regression model, we could illustrate the influence or “scale” of the determinants of poverty, which cannot be achieved by purely descriptive analysis [11,12]. We then compared the mortality rate during hospitalization between the low-income and the nonlow-income groups.

## 3. Results

Data from 4,647,058 injury inpatients between 1998 and 2015 were collected. The patient characteristics are summarized in Table 1. In all, 74,337 inpatients (1.6%) were low-income, and 4,572,721 (98.4%) were nonlow-income. The male-to-female ratio for the low-income group (1.74) was significantly higher than that of the nonlow-income group (1.41). The highest hospitalization rates occurred in patients aged 65 years and older in both the low-income group and the nonlow-income group. Hospitalization rates were also significantly different between the two groups for the 5–14 and 25–44 age groups. Low-income inpatients scored higher on the CCI measure than nonlow-income inpatients (0.6 and 0.5, respectively), indicating that the number and severity of injury complications were much higher for low-income inpatients.

Patients intentionally injured themselves at a higher rate in the low-income group (5.4%) than in the nonlow-income group (4.2%). Certain unintentional incidents, such as medical malpractice injuries and falls, were also more frequent in the low-income group (13.0% vs. 10.9% and 26.5% vs. 23.7%, respectively), whereas transport-related injuries and crushing, cutting, and piercing injuries were more frequent in the nonlow-income group (29.9% vs. 38.5% and 3.8% vs. 5.5%, respectively). In nonfatal cases, the two groups also differed significantly in medical malpractice injuries; falls; transport-related injuries; and crushing, cutting, and piercing injuries. Fatal injuries differed significantly by group only for transport-related injuries (20.9% vs. 27.9%) (Table 2).

Table 3 shows the distribution of treatment outcomes for unintentional and intentional injuries by group. Low-income inpatients were less likely than nonlow-income inpatients to receive surgery for unintentional and intentional injuries (47.4% vs. 55.2% and 28.2% vs. 30.4%, respectively). Fewer low-income inpatients sought treatment in medical centers for unintentional and intentional injuries (19.5% vs. 25.7% and 22.8% vs. 24.5%, respectively). More low-income inpatients who sought treatment in eastern Taiwan tended to stay longer in the hospital and incurred higher medical expenses compared to nonlow-income inpatients. Low-income inpatients also had a higher mortality rate than nonlow-income inpatients for both unintentional and intentional injuries (3.1% vs. 1.7% and 4.4% vs. 3.1%, respectively).

Table 4 shows the distribution of treatment outcomes for nonfatal and fatal injuries by group. Low-income inpatients were less likely to receive surgery for nonfatal injuries (41.6% and 51.4%, respectively) compared to nonlow-income patients. Low-income inpatients were also less likely to receive medical care in major hospital centers (21.3% vs. 29.1%) and were more likely to receive medical attention in regional hospitals (44.8% vs. 41.6%) compared to their counterparts. Moreover, low-income inpatients tended to stay longer in the hospital and incur higher medical expenses (9.9 days vs. 7.6 days and USD $1681.5 vs. USD $1573.9, respectively) than non-low-income patients. The comparative results for fatal injuries between the low-income group and the nonlow-income group were consistent with those found in regard to nonfatal injuries.

To further examine the factors associated with the injury hospitalization mortality rate, income, demographics (gender, age, etc.), CCI scores, intentionality of injury, cause of injury, and other hospitalization medical care-related measures (surgical operation, level of care, hospitalization area, and medical care utilization) were transformed into ratios using a variable within each respective group as a reference (Table 5). The mortality rate during hospitalization for the low-income group was almost twice (adjusted OR 1.888 (1.766, 2.018, *p* < 0.001) that of the nonlow-income group, indicating a strong correlation between mortality rate and income level. Table 1 indicates that low-income inpatients scored higher on the CCI measure than nonlow-income inpatients (0.6 and 0.5, respectively), indicating that the number and severity of injury complications were much higher for low-income inpatients. Older adults (65 years and older) in the low-income group had a higher mortality rate with respect to intentional injuries and surgery.

## 4. Discussion

### 4.1. Health Inequality

Between 1998 and 2015, an average of 237,877 citizens were classified as low-income [13], and their injury hospitalization rate was 1.74 per 100. In comparison, an average of 22,611,119 were classified as nonlow-income [14], and their injury hospitalization rate was 1.12 in 100. The injury hospitalization rate for the low-income group was twice as high as that of the nonlow-income group, implying that poverty and injury are correlated. In addition, low-income inpatients had more complicated injuries than nonlow-income inpatients (2.3 vs. 1.9), and the hospitalization mortality rate for low-income inpatients was 1.888 times higher than that of nonlow-income inpatients, showing that health inequality exists between the low-income and nonlow-income groups. The high hospitalization mortality rate for low-income inpatients is primarily driven by intentional injuries, as it was 2.014 times higher for nonlow-income inpatients. Poor people may easily feel desperate and have higher death rates due to “medical malpractice,” leading to health inequality. The findings of A. Case et al. are consistent with those of our study [15].

Many researchers have attributed the more complicated and severe injuries found in low-income individuals to their lower socioeconomic status, which may compel them to accept high-risk, entry-level jobs that require heavy labor. Consequently, these individuals are more susceptible to injuries, which may develop into severe and chronic conditions if left untreated. This is especially true for those who must work to support their families regardless of illness or injuries. These individuals are at higher risk of harming themselves due to their poor physical condition [16,17].

Serious injuries require more comprehensive care at advanced medical facilities. Our results showed that a significantly lower proportion of low-income inpatients received treatment in medical centers compared to nonlow-income inpatients, providing additional evidence of health inequality. Figure 1 illustrates differences between low-income inpatients and nonlow-income inpatients in the hospitalization payment processes under Taiwan’s NHI program, where the cost of a doctor visit includes medical expenses plus a registration fee, which is a processing or administration fee set by the medical institution that typically corresponds to the level of care. Therefore, the registration fees at medical centers are higher than those at regional and local hospitals. Additionally, while Taiwan’s NHI program covers most medical expenses, inpatients are still required to pay a small portion of their medical expenses as a copayment. To reduce health inequity, low-income inpatients are exempt from copayment. However, the NHI program does not cover the registration fees. Therefore, unless the local government or hospital social welfare measures (SWM) provide relief for low-income inpatients, these individuals have to pay the registration fees. In addition, postoperative patients may also be required to pay living costs and caregiver expenses during hospitalization and/or pay for other medical equipment or prescription drug expenses that are not covered by the NHI system. In general, nonlow-income inpatients have private health insurance to help pay those costs; however, low-income inpatients can only rely on minimum support from the government’s SWM, since they cannot afford private health insurance. Low-income inpatients’ hospitalization costs also tend to be higher than those of nonlow-income inpatients because they are hospitalized longer and are more seriously injured (Appendix A). Thus, low-income inpatients are more reluctant to receive surgery despite having higher CCI scores. Moreover, out-of-pocket medical expenses are generally proportional to the level of care received; the higher the level of care is, the greater the expense is. In Taiwan, the highest level of medical certification accreditation is the medical center. Consequently, low-income inpatients generally lean toward regional or local hospitals because of the lower level of expenses. However, when taking the severity of the injury into consideration, low-income inpatients are more willing to receive care at medical centers for fatal injuries as opposed to nonfatal injuries. The poor always spend money on expensive treatments instead of cheap prevention measures, which is consistent with our research results [18].

We also observed a significant gender gap in the low-income group. The Ministry of Health and Welfare of Taiwan has reported that among low-income individuals, the number of single-person households is higher for men than for women [19]. Typically, men in Taiwan are the main source of income because of the patriarchal nature of Chinese families. Therefore, low-income men are less likely to marry and more likely to accept jobs with poor working environments, thereby making them more vulnerable to injuries. In addition to the gender gap, the low-income and nonlow-income groups differed significantly in the percentage of individuals in the 5 to 14 age group (9.3% vs. 5.1%). Our findings were consistent with those of previous research, reporting that adolescents from lower socioeconomic families are more likely to have serious injuries requiring hospitalization [20]. Furthermore, poverty is more than just being low-income (although income is a highly significant determinant). Poor people are often stigmatized and feel ashamed, which may lead to self-isolation as a significant reason for their getting sick and as a primary reason for their avoidance of active, preventive healthcare.

### 4.2. Cause of Injury

Regarding unintentional, nonfatal injuries, our results showed that hospitalization rates related to “medical malpractice” injuries and “falls” were higher for the low-income group than for the nonlow-income group.

A multinational retrospective study on the global burden of disease (GBD) showed that, as a result of medical adverse events, disability-adjusted life years (DALYs) total 23 million globally every year, of which two-thirds come from low-income and mid-income countries [21]. Other research in these same countries has also shown more medical accidents, lower levels of patient safety, and lower levels of medical care quality for the low-income group [22].

We further observed that fall-related injuries were more prevalent among low-income inpatients in the middle-aged and elderly groups (more than 45 years old), as shown in Appendix A. Past research on the fall risk in older adults found that having a low income is a contributing factor [18]. Other factors include one’s socioeconomic status (low education, solitary living, and lack of care), living environment (inconvenient floorplan and insufficient light), and physical condition (poor vision, chronic illness, and aging) [23,24,25]. Accordingly, the low-income group may be more susceptible to falling and hospitalization because of their poor living conditions, the increased number of hazards in their environments, and their lack of safety equipment [24]. Intrinsic factors for falls include old age, muscle weakness, gait or balance problems, visual impairment, mobility problems, and cognitive impairment [26]. External factors related to falls include drugs, auxiliary devices, and hazardous environments (for example, uneven surfaces and poor lighting) [27]. These risk factors have been identified in previous studies.

Intentional injuries, i.e., “suicide” and “homicide,” showed a significantly higher rate of hospitalization for low-income inpatients in regard to both fatal and nonfatal injuries. Additionally, previous research has shown that unemployment and specific occupations are also associated with suicide and suicidal behavior [28,29,30,31]. According to a low-income family study conducted by the Ministry of the Interior of Taiwan, 62% of low-income family members have suffered catastrophic illnesses in the past [12]. In addition, 47% of the breadwinners in these families are out of the workforce.

Previous research has also found that poverty increases the risks for mental illness and suicide [32,33]. Low-income individuals generally view themselves as financial minorities and tend to feel powerless, helpless, and repressed when facing competition, which are factors that may be associated with contemplating suicide. Therefore, low-income individuals have a much higher risk of repeated suicide attempts that result in hospitalization [34]. Some studies have also indicated that low-income and severe illness exacerbate the risk of suicide during hospitalization. Patients with both severe illness and low income may suffer multiple complications that require long-term care. Without sufficient resources, these patients may become depressed and consider suicide to be an escape and relief for their families [35].

Duflo and Banerjee have personally participated in a number of aid programs that have aimed to penetrate the world of the poor in many countries and have investigated the 18 countries and regions with the highest concentrations of poor people; they have also used a large number of randomized controlled trials to recover people’s experiences, health, education, entrepreneurship, assistance, and many other aspects to explore the true roots of poverty [18]. Furthermore, one comprehensive analysis that collected criminal data from 169 countries found a positive correlation between income inequality and homicide/injury. This is especially prevalent in low-income and mid-income countries [36]. One could use Durkheim’s anomie theory [30] to explain this pattern. As someone tries to thrive in a society in which there are no legal avenues for achieving his or her goals, then he or she feels pressure to deviate from his or her usual behavior and engage in illegal activities. Similarly, lower social class members tend to turn their efforts to criminal activities because financial rewards in society are lacking, and criminal activities provide a means of obtaining financial rewards [37,38].

In contrast, nonlow-income inpatients were predominantly associated with transport-related injuries in both nonfatal and fatal injuries. According to official statistics [39], in 2010, 48.8% and 32.3% of traffic injuries were caused by motorcycles and private passenger cars, respectively. In Taiwan, motorcycles are the most popular means of transportation (64.1 motorcycles per 100 population) [40]. Most families have at least one motorcycle. Therefore, it is no surprise that our study found motorcyclist injuries to have the highest level of hospital admission rates for both the low-income and nonlow-income groups. As shown in Appendix A, motorcyclist injuries were more frequent in the low-income group than in the nonlow-income group. However, injuries suffered by the driver of a motor vehicle showed a more significant difference in hospital admission rates between the two groups compared to motorcyclist injuries. In general, low-income families are less likely to own a private passenger car because of the related costs or access barriers; therefore, they are less vulnerable to overall transport-related injuries. When the poor have the opportunity to obtain more income, they tend to buy more nutritious food that tastes better. Humans are creatures with strong desires, and it is this desire that allows us to defeat many of our competitors [41,42,43,44].

In summary, falling-related and transport-related injuries were the most common causes of injury in the low-income group. Typically, inpatients with injuries or older adults who are at an increased risk of falling often need to use mobility aids, such as wheelchairs, canes, and walkers. However, these mobility aids are not covered by Taiwan’s NHI, as per Article 51 of the National Health Insurance Act, and low-income inpatients are effectively denied access. Thus, health inequalities between low-income inpatients and nonlow-income inpatients still exist despite the implementation of Taiwan’s NHI. Government agencies should take actions to eliminate such health inequities for low-income patients.

### 4.3. Limitations

The data from the HWDC did not provide any information on immigration status (natives, new immigrants), marital status, education level, or occupation. The low-income group reported in this research was sorted according to those who qualified for insurance under the Public Assistance Act and not by actual income. The jobs of certain respondents, such as those who were unemployed, women, elderly individuals with a low pension, etc., were unmentioned. Health inequalities in the most vulnerable groups are likely to be higher than those in a low-middle income group (income quintiles or deciles). A Gini coefficient measuring health inequality was not estimated in the study. Therefore, potential nondifferential misclassification bias may exist and may have resulted in findings that favor the null hypothesis [45]. Therefore, this study may underestimate the differences between the low-income and the nonlow-income groups.

## 5. Conclusions

The low-income group had the highest hospitalization rates in patients aged 65 years or older, and the number and severity of injury complications were much higher for low-income inpatients. Low-income inpatients also had a higher mortality rate than nonlow-income inpatients for both unintentional and intentional injuries. The mortality rate during hospitalization for the low-income group was almost twice that of the nonlow-income group, indicating a strong correlation between mortality rate and income level. Low-income inpatients scored higher on the CCI measure than did nonlow-income inpatients, indicating that the number and severity of injury complications were much higher for low-income inpatients, thereby supporting our hypothesis that healthcare inequality exists and is correlated with income despite mandatory, nationwide, affordable medical insurance.

Therefore, the government should collaborate with private and nonprofit organizations to create a more comprehensive system, including employment assistance, job fairs, public education, etc., to gradually eliminate healthcare inequalities for low-income citizens. At the same time, we are committed to better health monitoring and analysis; advocating the integration of health into the governance aspects of interministerial committees; strengthening health-friendly policies, laws, and resource allocation from upstream; and paying attention to the special needs of different subgroups and the improvement of special health obstacles encountered, not only at the national level but also at the county, city, and service organization levels. Different groups (including those related to gender, age, ethnic group, region, social and economic status, persons with disabilities, people with different diseases, etc.) have special needs and resource utilization barriers, and therefore need the provision of health enhancements that are specifically formatted to overcome such barriers, services that are designed to minimize and prevent further barriers, and the ability to overcome barriers in an appropriate way that is effective at reducing health inequalities. In addition, transport- and fall-related injuries accounted for the majority of the injury cases in low-income inpatients. Therefore, as mobility aids promote national cross-departmental integrated fall prevention strategies, an integrated community health promotion model for the elderly population should be covered by the NHI program for low-income citizens to promote multiple intervention modes of fall prevention among elderly people of different ethnic groups to reduce the overall occurrence of falls in the elderly population.

## Figures and Tables

**Figure 1 healthcare-09-00349-f001:**
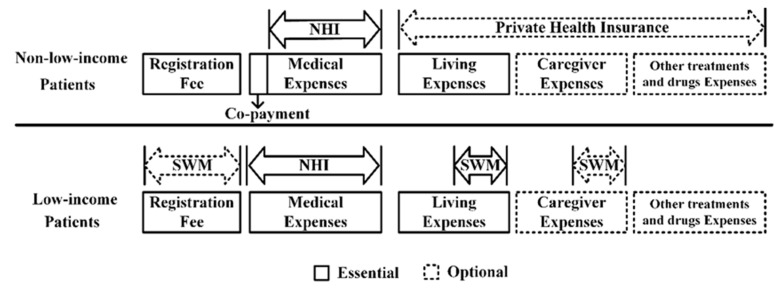
The comparison of medical costs between low-income inpatients and nonlow-income inpatients under the National Health Insurance (NHI) program in Taiwan.

**Table 1 healthcare-09-00349-t001:** Demographic variables of injury inpatients between 1998 and 2015 by income in Taiwan (*n* = 4,647,058).

Variables	Low-Income (*n* = 74,337)		Nonlow-Income (*n* = 4,572,721)		*p* Value
	N	%	N	%	
Gender					<0.001
Male	47,184	63.5	2,674,428	58.5	<0.001
Female	27,153	36.5	1,898,293	41.5	<0.001
Age					<0.001
1–4	1805	2.4	180,605	3.9	0.001
5–14	6902	9.3	231,217	5.1	<0.001
15–24	9816	13.2	657,796	14.4	0.090
25–44	15,739	21.2	1,154,947	25.3	<0.001
45–64	19,980	26.9	1,166,918	25.5	0.044
≥65	20,095	27.0	1,181,238	25.8	0.012
CCI	0.6 ± 1.5		0.5 ± 1.6		<0.001

*p* value: Chi-square/Fisher exact test on categorical variables, *t*-test on continuous variables, and percentage test on each item.

**Table 2 healthcare-09-00349-t002:** Causes of injury for inpatients of various incomes during 1998–2015 in Taiwan (*n* = 4,647,058).

Prognosis	Overall				*p* Value	Nonfatal (Survival)				*p* Value	Fatal (Mortality)				*p* Value
Incomes	Low-Income (*n* = 74,337)		Nonlow-Income (*n* = 4,572,721)			Low-Income (*n* = 71,640)		Nonlow-Income (*n* = 4,477,412)			Low-Income (*n* = 2697)		Nonlow-Income (*n* = 95,309)		
Causes of Injury	N	%	N	%		N	%	N	%		N	%	N	%	
Without E-Code (*n* = 1,582,244)	28,400		1,553,844			27,167		1,513,036			1233		40,808		
With E-Code (*n* = 3,064,814)	45,937		3,018,877			44,473		2,964,376			1464		54,501		
Unintentional	42,897	93.4	2,871,022	95.1	<0.001	41,566	93.5	2,821,399	95.2	<0.001	1331	90.9	49,683	91.2	0.173
Transport-related	13,723	29.9	1,162,189	38.5	<0.001	13,417	30.2	1,146,981	38.7	<0.001	306	20.9	15,208	27.9	<0.001
Poisoning	783	1.7	42,254	1.4	0.571	754	1.7	40,966	1.4	0.511	29	2.0	1288	2.4	0.785
Medical malpractice	5973	13.0	307,619	10.9	<0.001	5623	12.6	294,536	9.9	<0.001	350	23.9	13,083	24.0	0.896
Falls	12,184	26.5	714,467	23.7	<0.001	11,810	26.6	701,304	23.7	<0.001	374	25.5	13,163	24.2	0.771
Burns	201	0.4	9790	0.3	0.683	184	0.4	9505	0.3	0.981	17	1.2	285	0.5	0.483
Natural and environmental factors	451	1.0	29,904	1.0	0.997	446	1.0	29,768	1.0	0.995	5	0.3	136	0.2	0.982
Drowning	1221	2.7	166,939	5.5	0.408	1210	2.7	166,262	5.6	0.372	11	0.8	677	1.2	0.996
Suffocation	358	0.8	18,964	0.6	0.703	317	0.7	17,627	0.6	0.783	41	2.8	1337	2.5	0.934
Crushing, cutting, and piercing	1763	3.8	165,598	5.5	<0.001	1744	3.9	164,976	5.6	<0.001	19	1.3	622	1.1	0.890
Others unintentional	6240	13.6	253,298	8.4	<0.001	6061	13.6	249,414	8.4	<0.001	179	12.2	3884	7.1	0.061
Intentional	2479	5.4	125,816	4.2	0.001	2371	5.3	121,942	4.1	0.196	108	7.4	3874	7.1	0.042
Suicide	1137	2.5	52,533	1.7	0.029	1049	2.4	49,173	1.7	0.185	88	6.4	3360	6.1	0.047
Homicide	1342	2.9	73,283	2.4	0.040	1322	3.0	72,769	0.7	0.036	20	1.0	514	1.0	0.975
Unspecific and unable to determined	561	1.2	22,039	0.7	0.262	536	1.2	21,095	0.7	0.124	25	1.7	944	1.7	0.916

*p* value: percentage test.

**Table 3 healthcare-09-00349-t003:** Unintentional and intentional injury hospitalization-related variables for various incomes during 1998–2015 in Taiwan (*n* = 4,647,058).

Causes of Injury	Overall				*p* Value	Unintentional				*p* Value	Intentional				*p* Value
Incomes	Low-Income (*n* = 74,337)		Nonlow-Income (*n* = 4,572,721)			Low-Income (*n* = 42,897)		Nonlow-Income (*n* = 2,871,022)			Low-Income (*n* = 2479)		Nonlow-Income (*n* = 125,816)		
Variables	N	%	N	%		N	%	N	%		N	%	N	%	
Surgical operation					<0.001					<0.001					0.001
Yes	30,930	41.6	2,351,342	51.4		20,338	47.4	1,584,918	55.2		699	28.2	68,243	30.4	
No	43,407	58.4	2,221,379	48.6		22,559	52.6	1,286,104	44.8		1780	71.8	87,573	69.6	
Level of care					<0.001					<0.001					0.005
Medical center	15,804	21.3	1,330,952	29.1		8352	19.5	737,443	25.7		566	22.8	30,812	24.5	
Regional hospital	33,283	44.8	1,900,118	41.6		21,786	50.8	1,379,969	48.1		1165	47.0	58,342	46.4	
Local hospital	25,250	34.0	1,341,651	29.3		12,759	29.7	753,610	26.2		748	30.2	36,662	29.1	
Hospitalization area					<0.001					<0.001					<0.001
Northern	24,020	32.3	1,628,638	35.6		12,784	29.8	858,501	29.9		768	31.0	38,818	30.9	
Central	18,358	24.7	1,406,942	30.8		12,654	29.5	1,063,826	37.1		686	27.7	44,496	35.4	
Southern	21,881	29.4	1,239,379	27.1		12,056	28.1	759,659	26.5		716	28.9	33,892	26.9	
Eastern	9359	12.6	273,914	6.0		5034	11.7	178,115	6.2		298	12.0	8380	6.7	
Outer islands	719	1.0	23,848	0.5		369	0.9	10,921	0.4		11	0.4	230	0.2	
Medical care utilization															
Length of stays (day)	9.9 ± 11.5		7.6 ± 8.9		<0.001	9.1 ± 10.4		7.1 ± 7.9		<0.001	7.2 ± 9.6		5.4 ± 7.0		<0.001
Medical expenses (USD)	1681.5 ± 2880.3		1573.9 ± 2873.5		<0.001	1638.9 ± 2742.3		1484.8 ±2592.6		<0.001	1179.3 ± 2248.1		1068.5 ± 2294.9		<0.001
Prognosis					<0.001					<0.001					<0.001
Survival	71,640	96.4	4,477,412	97.9		41,566	96.9	2,821,399	98.3		2371	95.6	121,942	96.9	
Mortality	2697	3.6	95,309	2.1		1331	3.1	49,683	1.7		108	4.4	3874	3.1	

*p* value: Chi-square/Fisher exact test on categorical variables and *t*-test on continuous variables.

**Table 4 healthcare-09-00349-t004:** Fatal and nonfatal injury hospitalization-related variables by income level between 1998 and 2015 in Taiwan (*n* = 4,647,058).

Prognosis	Overall				*p* Value	Nonfatal (Survival)				*p* Value	Fatal (Mortality)				*p* Value
Incomes	Low-Income (*n* = 74,337)		Nonlow-Income (*n* = 4,572,721)			Low-Income (*n* = 71,640)		Nonlow-Income (*n* = 4,477,412)			Low-Income (*n* = 2697)		Nonlow-Income (*n* = 95,309)		
Variables	N	%	N	%		N	%	N	%		N	%	N	%	
Surgical operation					<0.001					<0.001					0.002
Yes	30,930	41.6	2,351,342	51.4		41,699	58.2	2,163,734	48.3		1708	63.3	57,645	64.1	
No	43,407	58.4	2,221,379	48.6		29,941	41.8	2,313,678	51.7		989	36.7	37,664	35.9	
Level of care					<0.001					<0.001					<0.00
Medical center	15,804	21.3	1,330,952	29.1		15,127	21.1	1,291,532	28.8		677	25.1	39,420	41.4	
Regional hospital	33,283	44.8	1,900,118	41.6		31,987	44.6	1,859,108	41.5		1296	48.1	41,010	43.0	
Local hospital	25,250	34.0	1,341,651	29.3		24,526	34.2	1,326,772	29.6		724	26.8	14,879	15.6	
Hospitalization area					<0.001					<0.001					<0.001
Northern	24,020	32.3	1,628,638	35.6		22,906	32.0	1,588,596	35.5		1114	41.3	40,042	42.0	
Central	18,358	24.7	1,406,942	30.8		17,798	24.8	1,381,480	30.9		560	20.8	25,462	26.7	
Southern	21,881	29.4	1,239,379	27.1		21,151	29.5	1,215,061	27.1		730	27.1	24,318	25.5	
Eastern	9359	12.6	273,914	6.0		9081	12.7	268,672	6.0		278	10.3	5242	5.5	
Outer islands	719	1.0	23,848	0.5		704	1.0	23,603	0.5		15	0.6	245	0.3	
Medical care utilization															
Length of stays (day)	9.9 ± 11.5		7.6 ± 8.9		<0.001	9.7 ± 11.3		7.5 ± 8.7		<0.001	13.8 ± 15.9		13.1 ± 15.4		<0.001
Medical expenses (USD)	1681.5 ± 2880.3		1573.9 ± 2873.5		<0.001	1551.2 ± 2520.1		1480.3 ±2565.4		<0.001	5968.6 ± 8201.8		5143.0 ± 6896.0		<0.001

*p* value: Chi-square/Fisher exact test on categorical variables and *t*-test on continuous variables.

**Table 5 healthcare-09-00349-t005:** Factors influencing mortality rate after hospitalization due to injury between 1998 and 2015.

	Overall			Low-Income		
Variables	Adjusted OR	95% CI	*p* Value	Adjusted OR	95% CI	*p* Value
Incomes						
Nonlow-income	Reference			-	-	-
Low-income	1.888	1.766–2.018	<0.001	-	-	-
Gender						
Male	1.482	1.451–1.520	<0.001	1.284	1.211–1.379	<0.001
Female	Reference			Reference		
Age						
1–4	Reference			Reference		
5–14	0.501	0.431–0.569	<0.001	0.767	0.620–0.885	0.001
15–24	1.083	0.980–1.186	0.088	1.120	1.013–1.271	0.032
25–44	1.421	1.318–1.554	<0.001	1.338	1.291–1.498	<0.001
45–64	2.125	1.973–2.339	<0.001	2.231	1.997–2.364	<0.001
≥65	5.076	4.452–5.561	<0.001	2.695	2.372–3.196	<0.001
CCI	1.142	1.133–1.142	<0.001	3.299	3.218–3.571	<0.001

Adjusted OR (odds ratio): adjusted variables listed in the table; CI = confidence interval. Hospitalization area had collinearity with level of care, and medical expenses had collinearity with length of stay; Nagelkerke *R*^2^ = 0.271 (overall), 0.234 (low income).

## Data Availability

Data are available from the NHIRD published by the Taiwan NHI administration. Because of legal restrictions imposed by the government of Taiwan concerning the “Personal Information Protection Act,” data cannot be made publicly available. Requests for data can be sent as a formal proposal to the NHIRD (http://www.mohw.gov.tw/cht/DOS/DM1.aspx?f_list_no=812).

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
