# Peer review of "Inequality in Health: The Correlation between Poverty and Injury—A Comprehensive Analysis Based on Income Level in Taiwan: A Cross-Sectional Study"

_healthcare, 2021, doi:10.3390/healthcare9030349_

Round 1
Reviewer 1 Report
The authors did a great job in taking my comments seriuosly into account. Thus, the paper improved significantly.
In a nutshell: the paper deals with a topic that is interesting from a methodological perspective (mostly with respect to the obstacles that may rise in using the chosen methods) but less so from a theoretical perspective. It incorporates theoretical and ethical problems of health, poverty and inequality marginally, although relevant to understand the analytical results. Taking the main purpose of the article into account, a publication in the present form is, however, well justified.
Author Response
Point 1: The authors did a great job in taking my comments seriously into account. Thus, the paper improved significantly.
In a nutshell: the paper deals with a topic that is interesting from a methodological perspective (mostly with respect to the obstacles that may rise in using the chosen methods) but less so from a theoretical perspective. It incorporates theoretical and ethical problems of health, poverty and inequality marginally, although relevant to understand the analytical results. Taking the main purpose of the article into account, a publication in the present form is, however, well justified.
Response 1: Dear Reviewer, thank you for the kind response. We have revised the manuscript according to your comments. We will consider our manuscript from a theoretical perspective in future study. We value your advice very much. Thank you.

Reviewer 2 Report
No further comments. The paper has improved significantly. The authors have incorporated my earlier comments.
Author Response
Point 1: No further comments. The paper has improved significantly. The authors have incorporated my earlier comments.
Response 1: Dear Reviewer, thank you for the kind response. We have incorporated your earlier comments in the manuscript. We value your advice very much. Thank you.

Reviewer 3 Report
Some changes and integrations have done by the authors but they aren't sufficient to resolve the methodological lack of the paper. The authors don’t have a clear hypothesis to test and all the study is confused in the description and analysis. The model used is inappropriate. The authors use data span 1998 to 2015 but the time variable is not considered at all, so the model can’t take into account the time correlation of data, collinearity and results are inefficient. The inference on estimated parameters are not valid. In conclusion, the analysis is inappropriate and invalidates all the results.
Author Response
Point 1: Some changes and integrations have done by the authors but they aren't sufficient to resolve the methodological lack of the paper.
Response 1: Dear Reviewer, thank you for the kind response. We have supplemented Table S6 to resolve the methodological lack of the paper.
The significance of the Hosmer and Lemeshow verification tests did not reach the significant level, indicating that the overall regression model is well-fitted. This also shows that the independent variable can effectively explain (and predict) the dependent variable. (Archer, Lemeshow, & Hosmer, 2007).
Collinarity: When two (or more) independent variables are not independent of each other (that is, related to each other), they have "collinarity". "Collinearity" will cause repeated independent variables in the regression model, improve the explanatory power and predictive power of a certain independent variable, and make the construction of the theory incorrect.( https://www.yongxi-stat.com/multiple-regression-analysis/)
The Variance Inflation Factor (VIF) measures the severity of multicollinearity in regression analysis. It is a statistical concept that indicates the increase in the variance of a regression coefficient as a result of collinearity.( https://www.yongxi-stat.com/multiple-regression-analysis/)
The VIF value is not greater than 10, it is judged that the collinearity between the independent variables is not serious, and the regression model can effectively predict the dependent variables. (https://www.yongxi-stat.com/multiple-regression-analysis/)
The VIF value of all variables in Table S6 is not greater than 10, it is judged that the collinearity between the independent variables is not serious, and the regression model can effectively predict the dependent variable.
We value your advice very much. Thank you.
Table S6. Influencing factors of injury hospitalization mortality rate between 1998 and 2015.
|
|
Overall |
|
Low-income |
||||||
|
Variables |
Adjusted OR |
95% CI |
P value |
VIF |
|
Adjusted OR |
95% CI |
P value |
VIF |
|
Incomes |
|
|
|
|
|
|
|
|
|
|
Nonlow-income |
Reference |
|
|
|
|
- |
- |
- |
- |
|
Low-income |
1.646 |
1.577-1.719 |
<0.001 |
1.002 |
|
- |
- |
- |
- |
|
Gender |
|
|
|
|
|
|
|
|
|
|
Male |
1.578 |
1.555-1.601 |
<0.001 |
1.021 |
|
1.419 |
1.290-1.560 |
<0.001 |
1.021 |
|
Female |
Reference |
|
|
|
|
Reference |
|
|
|
|
Age |
|
|
|
|
|
|
|
|
|
|
1-4 |
Reference |
|
|
|
|
Reference |
|
|
|
|
5-14 |
0.498 |
0.454-0.546 |
<0.001 |
2.188 |
|
0.440 |
0.241-0.804 |
0.008 |
4.416 |
|
15-24 |
1.007 |
0.943-1.075 |
0.841 |
3.992 |
|
0.834 |
0.491-1.418 |
0.503 |
5.591 |
|
25-44 |
1.411 |
1.327-1.500 |
<0.001 |
5.549 |
|
1.952 |
1.191-3.198 |
<0.001 |
6.690 |
|
45-64 |
2.267 |
2.135-2.408 |
<0.001 |
5.615 |
|
3.176 |
1.952-5.167 |
<0.001 |
7.938 |
|
≧65 |
5.355 |
5.047-5.681 |
<0.001 |
5.699 |
|
5.487 |
3.380-8.906 |
<0.001 |
7.936 |
|
CCI |
1.165 |
1.162-1.167 |
<0.001 |
1.061 |
|
1.161 |
1.143-1.179 |
<0.001 |
1.061 |
|
Year |
1.037 |
1.036-1.039 |
<0.001 |
1.014 |
|
0.998 |
0.989-1.007 |
0.622 |
1.016 |
Hosmer and Lemeshow test, P = 0.279 (Overall), 0.457 (Low-income), P > 0.05
Point 2: The authors don’t have a clear hypothesis to test and all the study is confused in the description and analysis.
Response 2: Dear Reviewer, thank you for the kind response. We have modified our hypothesis to test as follow Page 3. Line 108: There is a correlation between poverty and injury, which results in health inequality in low-income group.
We value your advice very much. Thank you.
Point 3: The model used is inappropriate.
Response 3: Dear Reviewer, thank you for the kind response. We have supplemented Table S6 to explain our model. The significance of the Hosmer and Lemeshow verification tests did not reach the significant level, indicating that the overall regression model is well-fitted. This also shows that the independent variable can effectively explain (and predict) the dependent variable. Hosmer and Lemeshow test, P = 0.279 (Overall), 0.457 (Low-income). Hosmer and Lemeshow test indicated that the model used is appropriate.
We value your advice very much. Thank you.
Table S6. Influencing factors of injury hospitalization mortality rate between 1998 and 2015.
|
|
Overall |
|
Low-income |
||||||
|
Variables |
Adjusted OR |
95% CI |
P value |
VIF |
|
Adjusted OR |
95% CI |
P value |
VIF |
|
Incomes |
|
|
|
|
|
|
|
|
|
|
Nonlow-income |
Reference |
|
|
|
|
- |
- |
- |
- |
|
Low-income |
1.646 |
1.577-1.719 |
<0.001 |
1.002 |
|
- |
- |
- |
- |
|
Gender |
|
|
|
|
|
|
|
|
|
|
Male |
1.578 |
1.555-1.601 |
<0.001 |
1.021 |
|
1.419 |
1.290-1.560 |
<0.001 |
1.021 |
|
Female |
Reference |
|
|
|
|
Reference |
|
|
|
|
Age |
|
|
|
|
|
|
|
|
|
|
1-4 |
Reference |
|
|
|
|
Reference |
|
|
|
|
5-14 |
0.498 |
0.454-0.546 |
<0.001 |
2.188 |
|
0.440 |
0.241-0.804 |
0.008 |
4.416 |
|
15-24 |
1.007 |
0.943-1.075 |
0.841 |
3.992 |
|
0.834 |
0.491-1.418 |
0.503 |
5.591 |
|
25-44 |
1.411 |
1.327-1.500 |
<0.001 |
5.549 |
|
1.952 |
1.191-3.198 |
<0.001 |
6.690 |
|
45-64 |
2.267 |
2.135-2.408 |
<0.001 |
5.615 |
|
3.176 |
1.952-5.167 |
<0.001 |
7.938 |
|
≧65 |
5.355 |
5.047-5.681 |
<0.001 |
5.699 |
|
5.487 |
3.380-8.906 |
<0.001 |
7.936 |
|
CCI |
1.165 |
1.162-1.167 |
<0.001 |
1.061 |
|
1.161 |
1.143-1.179 |
<0.001 |
1.061 |
|
Year |
1.037 |
1.036-1.039 |
<0.001 |
1.014 |
|
0.998 |
0.989-1.007 |
0.622 |
1.016 |
Hosmer and Lemeshow test, P = 0.279 (Overall), 0.457 (Low-income), P > 0.05
Point 4: The authors use data span 1998 to 2015 but the time variable is not considered at all, so the model can’t take into account the time correlation of data, collinearity and results are inefficient.
Response 4: Dear Reviewer, thank you for the kind response. We have supplemented Table S6 to explain time correlation of data, collinearity and results are efficient.
We have considered the time variable in Table S6 and the result of variable Year (Over all) Adjusted OR is 1.037, variable Year (Low-income) Adjusted OR is 0.998. Variable Year can explain time correlation of data.
The VIF value is not greater than 10, so it is judged that the collinearity between the independent variables is not serious, and the regression model can effectively predict the dependent variables.
According to the variable Year and VIF value, the results are efficient.
We value your advice very much. Thank you.
Table S6. Influencing factors of injury hospitalization mortality rate between 1998 and 2015.
|
|
Overall |
|
Low-income |
||||||
|
Variables |
Adjusted OR |
95% CI |
P value |
VIF |
|
Adjusted OR |
95% CI |
P value |
VIF |
|
Incomes |
|
|
|
|
|
|
|
|
|
|
Nonlow-income |
Reference |
|
|
|
|
- |
- |
- |
- |
|
Low-income |
1.646 |
1.577-1.719 |
<0.001 |
1.002 |
|
- |
- |
- |
- |
|
Gender |
|
|
|
|
|
|
|
|
|
|
Male |
1.578 |
1.555-1.601 |
<0.001 |
1.021 |
|
1.419 |
1.290-1.560 |
<0.001 |
1.021 |
|
Female |
Reference |
|
|
|
|
Reference |
|
|
|
|
Age |
|
|
|
|
|
|
|
|
|
|
1-4 |
Reference |
|
|
|
|
Reference |
|
|
|
|
5-14 |
0.498 |
0.454-0.546 |
<0.001 |
2.188 |
|
0.440 |
0.241-0.804 |
0.008 |
4.416 |
|
15-24 |
1.007 |
0.943-1.075 |
0.841 |
3.992 |
|
0.834 |
0.491-1.418 |
0.503 |
5.591 |
|
25-44 |
1.411 |
1.327-1.500 |
<0.001 |
5.549 |
|
1.952 |
1.191-3.198 |
<0.001 |
6.690 |
|
45-64 |
2.267 |
2.135-2.408 |
<0.001 |
5.615 |
|
3.176 |
1.952-5.167 |
<0.001 |
7.938 |
|
≧65 |
5.355 |
5.047-5.681 |
<0.001 |
5.699 |
|
5.487 |
3.380-8.906 |
<0.001 |
7.936 |
|
CCI |
1.165 |
1.162-1.167 |
<0.001 |
1.061 |
|
1.161 |
1.143-1.179 |
<0.001 |
1.061 |
|
Year |
1.037 |
1.036-1.039 |
<0.001 |
1.014 |
|
0.998 |
0.989-1.007 |
0.622 |
1.016 |
Hosmer and Lemeshow test, P = 0.279 (Overall), 0.457 (Low-income), P > 0.05
Point 5: The inference on estimated parameters are not valid. In conclusion, the analysis is inappropriate and invalidates all the results.
Response 5: Dear Reviewer, thank you for the kind response. We have supplemented Table S6 to explain time correlation of data, collinearity and results are efficient. The inferences on the estimated parameters are valid. The analysis is appropriate and verifies all results.
We value your advice very much. Thank you.
Table S6. Influencing factors of injury hospitalization mortality rate between 1998 and 2015.
|
|
Overall |
|
Low-income |
||||||
|
Variables |
Adjusted OR |
95% CI |
P value |
VIF |
|
Adjusted OR |
95% CI |
P value |
VIF |
|
Incomes |
|
|
|
|
|
|
|
|
|
|
Nonlow-income |
Reference |
|
|
|
|
- |
- |
- |
- |
|
Low-income |
1.646 |
1.577-1.719 |
<0.001 |
1.002 |
|
- |
- |
- |
- |
|
Gender |
|
|
|
|
|
|
|
|
|
|
Male |
1.578 |
1.555-1.601 |
<0.001 |
1.021 |
|
1.419 |
1.290-1.560 |
<0.001 |
1.021 |
|
Female |
Reference |
|
|
|
|
Reference |
|
|
|
|
Age |
|
|
|
|
|
|
|
|
|
|
1-4 |
Reference |
|
|
|
|
Reference |
|
|
|
|
5-14 |
0.498 |
0.454-0.546 |
<0.001 |
2.188 |
|
0.440 |
0.241-0.804 |
0.008 |
4.416 |
|
15-24 |
1.007 |
0.943-1.075 |
0.841 |
3.992 |
|
0.834 |
0.491-1.418 |
0.503 |
5.591 |
|
25-44 |
1.411 |
1.327-1.500 |
<0.001 |
5.549 |
|
1.952 |
1.191-3.198 |
<0.001 |
6.690 |
|
45-64 |
2.267 |
2.135-2.408 |
<0.001 |
5.615 |
|
3.176 |
1.952-5.167 |
<0.001 |
7.938 |
|
≧65 |
5.355 |
5.047-5.681 |
<0.001 |
5.699 |
|
5.487 |
3.380-8.906 |
<0.001 |
7.936 |
|
CCI |
1.165 |
1.162-1.167 |
<0.001 |
1.061 |
|
1.161 |
1.143-1.179 |
<0.001 |
1.061 |
|
Year |
1.037 |
1.036-1.039 |
<0.001 |
1.014 |
|
0.998 |
0.989-1.007 |
0.622 |
1.016 |
Hosmer and Lemeshow test, P = 0.279 (Overall), 0.457 (Low-income), P > 0.05

Reviewer 4 Report
No suggestions for authors
Author Response
Point 1: No suggestions for authors.
Response 1: Dear Reviewer, thank you for the kind response. We value your advice very much. Thank you.

Round 2
Reviewer 3 Report
No further comments. The authors has taking my comments seriously into account and improved the paper significantly.
This manuscript is a resubmission of an earlier submission. The following is a list of the peer review reports and author responses from that submission.
Round 1
Reviewer 1 Report
The paper aims to evaluate correlations between income level (‘low-income’ vs ‘non-low-income’) and health inequality in general and poverty in particular. Correlations refer to descriptive statistics only without any other method (multivariate regression, discriminant analysis, etc.). Also, the paper focuses only on a methodological discussion of inequality, poverty and inpatients, and is thus missing a theoretical reflection on these core terms.
Using only a binary variable to distinguish poverty from non-poverty does not represent the complexity of the problem discussed in the article. Also, using only a threshold value does not allow for an analysis of ‘inequality’ as the paper is aiming for. For example, a Gini coefficient would be necessary at least to measure health inequality.
The choice of the selected cases of ‘injury causes’ is not justified, although it is not self-explaining. Almost all health care incidents refer to accidents, followed by suicide and homicide as well as natural and environmental factors. Poor people, however, suffer mostly from chronic diseases, stroke, cardiovascular disease, malnutrition, and obesity; furthermore, and not to forget, psychic problems due to many reasons (lack of resources and recognition). The critique thus is twofold: why does the paper not include poor-prone health problems? And why does it include accidents, homicide/suicide, which thematically have no apparent relation (correlation, association) to income-poor people?
Furthermore, poverty is more than just being income-poor (income is a highly significant determinant). Poor people are often stigmatized and feel ashamed, which may lead to self-isolation as a significant reason for getting sick and as a primary reason for avoiding active, preventive health care.
Page 3, lines 109-116: It may be statistically significant that variations occur of mentioned types of injuries and the two income groups. However, a content-related or objective significance cannot be assumed straightforwardly. What were the respective hypotheses? They are not mentioned nor explained.
Page 9, lines 5-7: the fact that poor people tend to have a higher mortality rate during hospitalization is – at the descriptive level – not surprising at all. Explanations are harder to obtain and refer to their health status, which in turn is influenced by access (and other causes) to health care institutions.
Chapter 4.1 Health inequality: this chapter does not really consider issues of inequality, but presents descriptive statistical results for two groups; there is no further differentiation which would allow for analyses of inequalities.
Page 10, lines 25-30: this explanation seems to be tempting but actually refers to only one subgroup of the low-income class. It leaves unmentioned unemployed people, women, the elderly with a low pension, etc.
Page 10, lines 31-56: this is a very good paragraph, highlighting some evident background information about different patterns of behaviors of the two income groups. These delineations are self-evident, i.e. statistics may add some quantitative confirmation but do not explain the facts.
Page 11, lines 87-93: again, the paper reports a descriptive statistical correlation and conclude that “low-income group may be more susceptible to falling …” but cannot confirm it. Furthermore, a binary differentiation of poverty oversimplifies the proportions of inpatients and hospitalization. A correct distribution of inequality of the entire population with particular attention of the lower-income range is missing.
The concluding chapter is much too short of addressing conclusions on what has been presented. The recommendations given in this chapter appear to be very general and do not represent a tailored response to the results shown in the previous chapters.
Reviewer 2 Report
This paper is the first that I know which makes the case that low-income people behave very differently. They take undue risk and make choices that people with high income will avoid because they face prohibitively high fixed cost.
Putting it differently, health (like education) is a part of human capital. There are increasing returns to scale from the fixed cost of investment in health and the complementarity of parents knowledge about health. These increasing returns lead to "poverty-trap" causing the low-income people's behaviour very different from others.
In "Poor Economics" (MIT Press, 2011), the 2019 Nobel Prize-winning economists Abhijit Banerjee and Esther Duflo make the case of poor people behaving differently very powerfully.
This paper provides valuable evidence to support that literature. The author should cite them and bolster the value of their contribution by presenting their findings as evidence to Banerjee and Duflo's hypothesis.
Here are a few key theoretical papers for which the findings of this paper may be quite relevant.
1. Banerjee, A. V. and Duflo, E. (2007) "The Economic Lives of the Poor," Journal of Economic Perspectives, 21, 141–167.
2. Moav, O. (2005). "Cheap Children and the Persistence of Poverty," The Economic Journal, 115, 88-110.
3. Mookherjee, D. Prina, S. and D. Ray (2012). "A Theory of Occupational Choice with Endogenous Fertility," American Economic Journal: Microeconomics 2012, 4, 1-34.
4. Rosenzweig, M. R. and K. I. Wolpin (1994). "Are There Increasing Returns to the Intergenerational Production of Human Capital? Maternal Schooling and Child Intellectual Achievement." Journal of Human Resources, 29(2), Special Issue: Women's Work, Wages, and Well-Being, 670-693.
Regarding the empirical methodology, some justification for the method used would be helpful. Authors could cite a published article in a high-quality journal that uses a similar method to draw an inference. Some discussion on the merit of this statistical procedure for deriving inference would improve the credibility of this evidence.
Reviewer 3 Report
This manuscript tried to highlights inequalities in health among income groups using Twaian NHI data. But I thought that several things should be improved to be published in the journal.
- low-income group accounts for 1.6% of total inpatients and the rest subject to non-low-income groups. This classification is very strange in health inequity studies most of which have used 5 or 3 group by income level. I' do not understand what are 'non-low-income groups'. Please explain why you used this classification in income level.
- The main hypothesis is whether poverty correlate to injury or not. Table 5 showed influencing factors to injury mortality. You should made a change table 5 to be suitabe for your hypothesis.
Reviewer 4 Report
REVIEW OF THE PAPER ON HELTHCARE
Inequality in Health: Correlation Between Poverty and Injury—A Comprehensive Analysis Based on Income Level in Taiwan: A Cross-sectional Study
The research focuses on the study of relationship between income and injury hospitalization mortality rate in Taiwan from 1998 to 2015, after the mandatory participation for all citizens a basic level of medical care of National Health Insurance in 1995.
The aim of the paper is not clear explained, the authors talking about mortality rate and then of inequality in health. To analyze the mortality rate is very different from the analysis of health inequality.
The study has different problems and the paper is not adequately developed.
The article presents a rather limited review of the literature. When the authors will clarify the main aim of the paper, deciding if to focus on inequality of health or in mortality rate, a greater effort is necessary to improve the analysis of previous literature.
The multivariate statistical model considered in the analysis is inappropriate. In the regression model the time factor is not considered at all. In order to appreciate the incidence of the temporal factor, it would be more appropriate to use a model for panel data, which is more informative and able to provide greater variability, less collinearity between the variables, a higher number of degrees of freedom and greater efficiency of the estimates.
The response variable is not appropriately considered because it take into consideration fatal event in very different morbidity cases that depend in particular from the job of the respondent that here isn’t considered.
Discussion of results must be improved, at this regard, it is necessary to explain the connection between the significant explanatory variables and the dependent variable and provide a comparison with other studies in the literature.
For all these reasons the paper can’t be considered for publication
Reviewer 5 Report
Dear authors,
Well written article on the important topic. Few simple comments:
- Page 2 L68: Please use a colon mark after the " ICD-9CM E-Code";
- Page 2 L70: Please use a colon mark after the "ICD-9-CM E-Code";
- P2 L81-82: Capitalize the name of the Central Department of Budget, Accounting and Statistics;
- P 11 L82-83: Capitalize the names of the GBD and DALY's in the sentence;
- P 11 L88 and P12 L121: Should be written "Table 2" and "Table 3" without "S"? Or do you mean something specific under the “Table S2" and "Table S3"?